# Modular coherent photonic-aided payload receiver for communications satellites

Vanessa C. Duarte [1,2], João G. Prata[1], Carlos F. Ribeiro[1], Rogério N. Nogueira[1,3], Georg Winzer[2], Lars Zimmermann[2], Rob Walker[4], Stephen Clements[4], Marta Filipowicz[5], Marek Napierała[5], Tomasz Nasiłowski[5], Jonathan Crabb[6], Marios Kechagias[6], Leontios Stampoulidis[6], Javad Anzalchi[7] & Miguel V. Drummond [1]

Ubiquitous satellite communications are in a leading position for bridging the digital divide. Fulfilling such a mission will require satellite services on par with fibre services, both in bandwidth and cost. Achieving such a performance requires a new generation of communications payloads powered by large-scale processors, enabling a dynamic allocation of hundreds of beams with a total capacity beyond 1 Tbit s$^{-1}$. The fact that the scale of the processor is proportional to the wavelength of its signals has made photonics a key technology for its implementation. However, one last challenge hinders the introduction of photonics: while large-scale processors demand a modular implementation, coherency among signals must be preserved using simple methods. Here, we demonstrate a coherent photonic-aided receiver meeting such demands. This work shows that a modular and coherent photonic-aided payload is feasible, making way to an extensive introduction of photonics in next generation communications satellites.

[1] Instituto de Telecomunicações, Campus Universitário de Santiago, 3810-193 Aveiro, Portugal. [2] IHP, Im Technologiepark 25, 15236 Frankfurt (Oder), Germany. [3] Watgrid Lda., Via do Conhecimento, 3830-352 Ílhavo, Portugal. [4] aXenic Ltd., Thomas Wright Way, Sedgefield TS21 3FD, UK. [5] InPhoTech Sp. z o.o., Meksykańska 6 lok. 102, Warsaw 03-948, Poland. [6] Gooch & Housego, Broomhill Way, Torquay TQ2 7QL, UK. [7] Airbus Defence & Space, Gunnels Wood Rd, Stevenage SG1 2AS, UK. Correspondence and requests for materials should be addressed to V.C.D. (email: vanessaduarte@av.it.pt)

The evolution of high-throughput communication satellites (HTCSs) has been paced by a raw increase in capacity, obtained by packing more beams within the coverage zone of the satellite[1]. So far, adding a new beam has been typically achieved by introducing at least one more feed to the payload, placing it according to the position of the beam[2]. While such a design paradigm stood for more than half a century, it has now become obsolete for two main reasons. First, it scales aggressively. The number of feeds increases not only in proportion to the number of beams but also inversely to their width, as thinner beams require larger apertures encompassing multiple feeds[3]. Second, user needs are changing at an accelerating rate. Having a static beam configuration matching user needs at all times during the entire lifetime of the satellite, typically spanning 15 years, is nowadays impossible. Consequently, satellite operators recently started asking for flexible satellite payloads[1,4], with satellite QUANTUM being the first example of a communications satellite designed with a programmable digital signal processor. QUANTUM offers a total bandwidth of 3.5 GHz which can be flexibly allocated to eight steerable spot beams. The corresponding capacity is almost 30 times lower than already launched HTCSs, presently beyond 100 Gbit s$^{-1}$, evidencing that capacity and flexibility could not be simultaneously increased with either radio frequency (RF) or digital technologies.

A scalable capacity increase and a flexible beam configuration can be both achieved by operating the feeds as a phased array antenna (PAA), as illustrated in Fig. 1. Without loss of generality, let us consider a PAA comprising $N$ feeds and receiving a set of $N_B$ beams. Each feed now receives all beams, each beam arriving with a unique delay with respect to other feeds. Such a property enables separating all $N_B$ beams from the $N$ signals produced by the PAA by multiplying these by a matrix with $N_B \times N$ coefficients. In practice, such a matrix is implemented by a processor denominated by beamforming network (BFN), and each coefficient is implemented by an amplifier or attenuator combined with a phase shifter. In terms of scale, the required number of feeds increases only inversely to the minimum beam width, resulting in the lowest possible number of feeds[3,5]. As for flexibility, a full-scale reconfigurable BFN is able to adapt to any set of beams.

While implementing the feeds to operate as a PAA is trivial, a full-scale implementation of a BFN comprising all $N_B \times N$ coefficients has so far been too cumbersome to implement either with digital or analogue signal processors. On the one hand, digitally processing all signals provided by the PAA requires an unrealistic processing power of at least 1 Gsa s$^{-1}$ per input signal and per beam, over 10 Tsa s$^{-1}$ for already launched HTCSs[1,3]. On the other hand, the size of an analogue BFN depends on the length of each phase shifter, which is proportional to the wavelength of the RF signals. This results in a deadlock: while a BFN processing high-frequency RF signals is inherently compact, a low-loss implementation is challenging due to the high frequency of the RF signals. Photonics allows overcoming such a trade-off. The wavelength of optical signals is more than 5000 times shorter than a typical Ka-band RF signal, enabling significant miniaturization of the BFN. Yet optical waveguides, either fibre or within a photonic integrated circuit (PIC), are well known for being low-loss[6]. Such unique advantage puts photonics in a leading position for implementing a BFN.

A photonic-aided payload implements the BFN with a programmable photonic processor, resulting in an optical beamforming network (OBFN)[5,7,8]. A miniaturizable OBFN must be identical to an RF BFN, with an optical phase shifter being equivalent to an RF phase shifter. Consequently, a miniaturizable OBFN relies on coherent optical signal processing, which in turn enables coherent detection[3,5,9,10]. Coherent detection can also provide heterodyning, thus enabling RF frequency conversion[3,5,11]. As a result, RF hardware is assigned only to basic tasks such as amplification and inverse/output multiplexing[3]. The main advantage of such an approach is that it does not change the payload architecture, allowing to keep mandatory function modularity and redundancy mechanisms[3]. However, the sheer scale of the photonic-aided payload unavoidably results in long optical paths, more than 5000 times longer than RF paths when normalized to the wavelength. As a result, unavoidable thermal and mechanical gradients imposed to optical paths produce a slow but random phase drift to the propagating signals, obliterating beamforming if no action is taken. Given that there is no such problem in an RF BFN, its configuration may be performed by a fairly static monitoring and control loop (MCL). Conversely, an OBFN requires a dynamic MCL, which nonetheless must be kept simple, scalable and fast just enough to

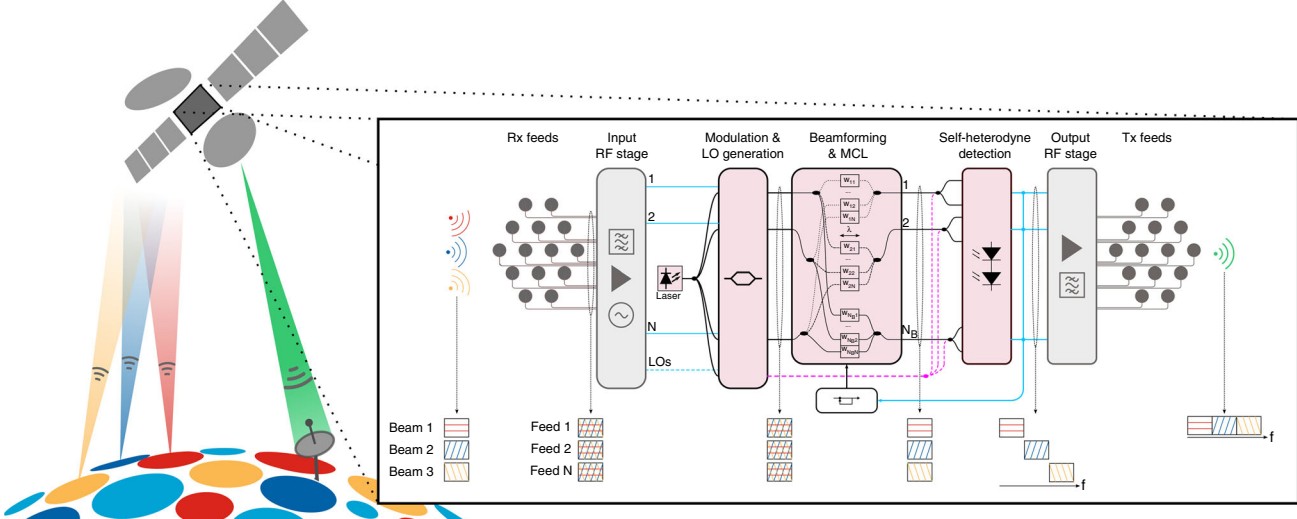

**Fig. 1** Photonic-aided payload receiver. User beams are received by the satellite, aggregated by the payload and re-transmitted to a ground station connected to the World Wide Web. Inset: the photonic-aided payload receiver up-converts the signals provided by the receiving feeds to optical signals, separates all beams via beamforming, and down-converts the separated beams back to radio frequency (RF) signals at the right frequency to be re-transmitted to a ground station by means of heterodyning. A monitoring and control loop (MCL) is used to operate, optimize and stabilize the optical beamforming network (OBFN)

track phase drifting, in order to be suitable for a large-scale OBFN. While all stages of a photonic-aided payload have been individually validated—modulation[6,12–16], beamforming[9,10,17–21], frequency conversion[22,23] and coherent detection[6,9,24]—such an MCL has never been demonstrated, preventing the demonstration of a scalable, modular and coherent photonic-aided payload.

In this paper, we demonstrate a complete scalable, modular and coherent photonic-aided payload receiver comprising four custom-made modules, all aiming for miniaturization and modularity: two arrays of two GaAs Mach–Zehnder modulators (MZMs) each, a radiation-hardened erbium-doped multicore fiber amplifier (EDMCFA) with seven cores, a 4-by-1 integrated OBFN, and an MCL for defining and stabilizing the amplitude, delay and phase of each signal. The receiver was fed with two beams arriving from different directions, each carrying a 1 Gbit s$^{-1}$ quadrature phase shift keying (QPSK) signal at 28 GHz, being able to separate one from the other in real time. To the best of our knowledge, it is the first demonstration of real-time beam separation ever achieved by a complete photonic-aided payload receiver.

## Results

**Self-coherent photonic processor.** The main task of a payload receiver is to down-convert the frequency of an input beam such that it can be processed by an inverse multiplexer[3]. Frequency down-conversion is accomplished by heterodyning the beam with a local oscillator (LO), which is typically derived from a single reference oscillator[3]. A photonic-assisted payload receiver should likewise resort to heterodyne detection, with optical local oscillators (OLOs) taken from a single reference OLO, thus resulting in self-heterodyne detection[5,20]. Self-heterodyne detection has the important advantage of cancelling laser phase noise, which relaxes the required laser linewidth, and makes the proposed MCL possible.

The self-coherent photonic processor at the core of the proposed photonic-assisted payload receiver complies with such principle by using a single laser source, as depicted in Fig. 2a. The processor first converts input RF signals to the optical domain. The converted signals are individually amplified, power equalized, phase shifted, delayed and coherently added into one output

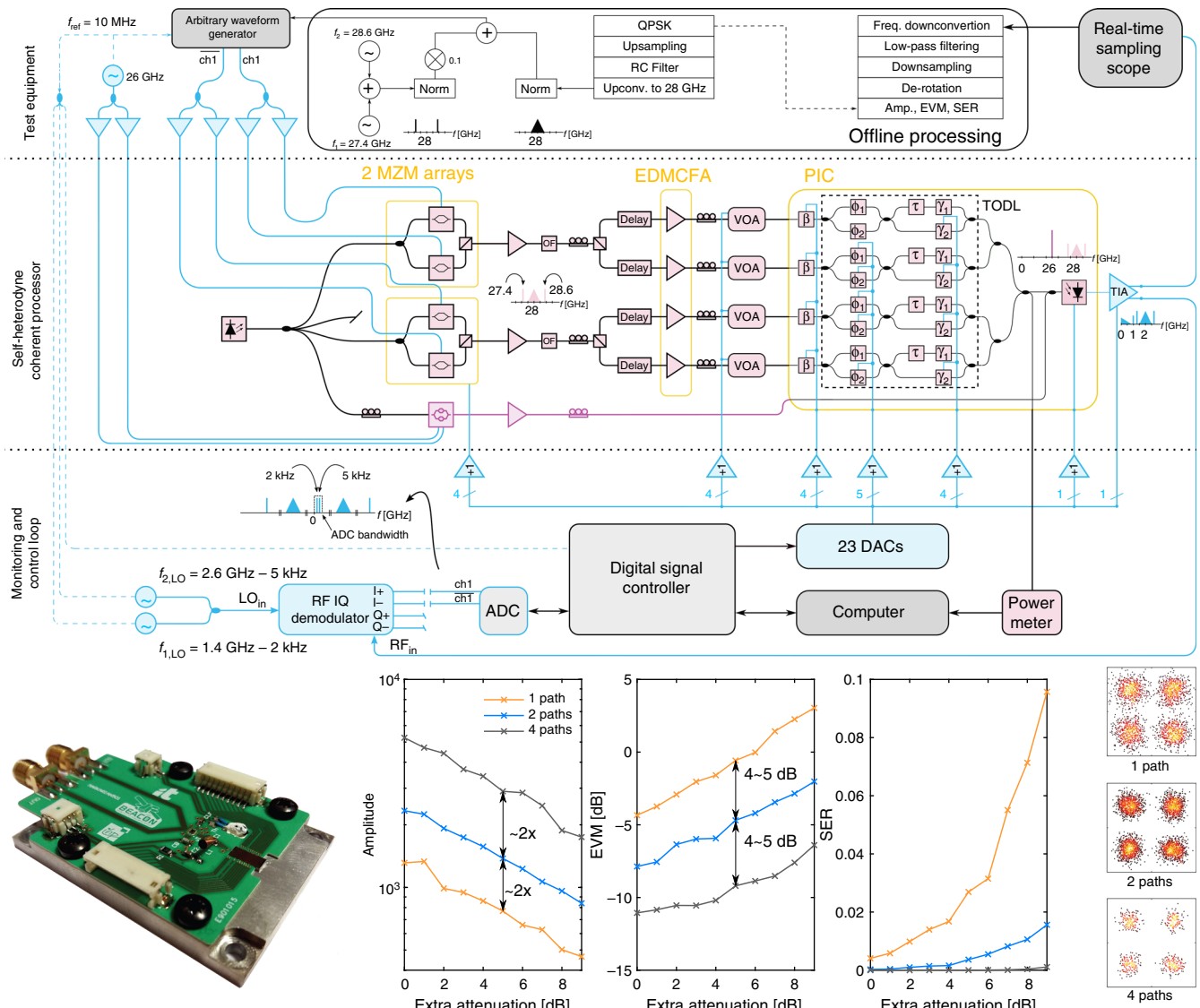

**Fig. 2** Set up and characterization results of the proposed photonic processor. **a** Experimental set-up. **b** The photonic integrated circuit (PIC) is bond-wired to a printed circuit board (PCB), which provides lateral direct current (DC) access to the phase shifters and to a thermistor, and two radio frequency (RF) connectors for accessing the differential outputs of the transimpedance amplifiers (TIA). The pads of the germanium photodiode (Ge-PD) are bond-wired to a bias tee, which interfaces the photodiode with the transimpedance amplifier (TIA). **c** Amplitude, error vector magnitude (EVM) and symbol error rate (SER) of the output signal obtained when adding one, two and all four signals. **d** Representative constellation diagrams of the output signal obtained without extra attenuation

optical signal. The resulting signal is coherently detected through heterodyning, therefore resulting in down-conversion of the RF frequency[20,25]. A low-complexity pilot-aided MCL is used to control and stabilize the system in real time. The experimental set-up is detailed as follows.

Light from a TLS (tunable laser source) is split by three paths. In the two upper paths, RF-to-optical conversion is achieved by means of amplitude modulation done by two arrays of GaAs MZMs, each with two MZMs[13,26]. Each array outputs the two modulated signal to a single fibre owingthanks to polarization multiplexing. The modulated signals are then pre-amplified, noise filtered and polarization demultiplexed. For each of the four resulting optical signals there is a dedicated free-space optical delay line to equalize the relative delays of the modulated signals within the tuning range of the tunable optical delay lines (TODLs) of the OBFN. The delay-equalized signals are then boosted by four of the seven cores of a saturated EDMCFA[27]. The amplified signals are equalized in power by variable optical attenuators (VOAs), and then fed to an integrated OBFN embedded in a printed circuit board (PCB), as shown in Fig. 2b. Details of the silicon PIC implementing the integrated OBFN are given in the Supplementary Note 4. The integrated OBFN comprises four identical paths, each with an input phase shifter, for adjusting the phase of each input signal, and a TODL based on a Mach–Zehnder delay interferometer (MZDI) with variable coupling ratio and with a tuning range of $\tau = 50$ ps[20]. The four delayed signals are combined into a processed output signal that, in turn, is split in two copies. One is routed to an external power metre, which is connected to the MCL. The other is combined with an OLO and fed to a Germanium photodiode (Ge-PD), thus being coherently detected. The resulting electrical signal is amplified by a transimpedance amplifier (TIA) with two differential outputs, one of which is the electrical output signal of the photonic processor. The OLO is generated in the purple-coloured optical path as follows. An I/Q modulator driven by two RF tones with a frequency of $f_{OLO}$ and dephased by $\pi/2$ produces a frequency-shifted version of the laser signal. The modulated signal is amplified and combined with the processed output signal, thus serving as a frequency-shifted optical local oscillator (FSOLO). Consequently, the processed signal is frequency down-converted by $f_{OLO}$. Further details about the experimental set-up can be found in the Supplementary Note 3.

An MCL is required for setting up and stabilizing the amplitude, phase and delay of each input optical signal. A simple, scalable and low-power MCL should rely only on the output signals of the processor, and handle as few RF signals as possible[25]. The proposed MCL relies on two out-of-band pilot tones with frequencies $f_1$ and $f_2$ added to the input RF signals, thus being transparent to the input RF signals and associated propagation impairments. As explained in ref. [28], using a pair of pilot tones enables a simple and precise estimation of the time delay of a given path, as the time delay is proportional to the phase difference between pilot tones. Amplitude and phase can be directly estimated from one of the pilot tones. Given that identical pilot tones are fed to all input RF signals, and that the MCL takes as input the output electrical signal, a method for distinguishing the pilot tones associated with different signals must be enforced[28]. A simple method is here proposed, in which weak dithering tones with different low frequencies (<2 kHz) are digitally generated and fed to the phase shifters $\beta$. As a result, the output signal comprises the RF signal, pilot tones $f_1$ and $f_2$, and weak dithering tones $f_1 \pm f_{d,k}$ and $f_2 \pm f_{d,k}$, where $f_{d,k}$ is the frequency of the dithering tone fed to the phase shifter $\beta$ of the path $k$. The MCL thus uses both dithering tones $f_1 \pm f_{d,k}$ and $f_2 \pm f_{d,k}$ to estimate the delay of path $k$, and at least one of such tones to estimate the amplitude and phase of the corresponding signal.

While the amplitude and time delay of the optical signals are fairly static, the phase wanders over time as a result of temperature and mechanical gradients affecting optical fibres. Consequently, the MCL refresh rate should be high enough to counteract phase wandering, thus providing fundamental stability when coherently combining optical signals in the OBFN. Such is the case of the presented set-up, as explained in the Supplementary Note 1. The impact caused by parasitic phase modulation on the signal is also discussed in the Supplementary Note 2.

The implementation of the MCL is depicted at the bottom of Fig. 2a. An RF I/Q demodulator is used to simultaneously down-convert the spectral content around $f_1$ to 2 kHz and around $f_2$ to 5 kHz. The resulting signal, containing all pilot and dithering tones at low frequency, is digitized by an analog-to-digital converter (ADC) with a low sampling rate. A digital signal controller (DSC) is used to control and synchronize the acquisition of samples by providing a time stamp, allowing accurate digital down-conversion of all dithering tones to baseband, and thus phase and delay estimation. Frequency down-conversion, parameter estimation and control are split between the DSC and a computer. The control algorithms output new voltages to be applied to the phase shifters and VOAs, which are programmed in the respective digital-to-analog converters (DACs) via the DSC. All components used for implementing the MCL are low-end commercial off-the-shelf.

We validated the operation of the proposed photonic processor by setting it up to coherently add four identical RF signals with equalized power. As shown in Fig. 2a, we programmed an arbitrary waveform generator (AWG) to generate a QPSK signal at 1 Gbit s$^{-1}$ comprising 640 symbols, pulse-shaped by a raised cosine (RC) filter with a roll-off factor of 0.25 GHz, with a carrier frequency of 28 GHz, and with two pilot tones at $28 \pm 0.6$ GHz each with an amplitude of 10% of the QPSK signal. Such a signal was repetitively generated. Four copies of the RF signal were produced using RF splitters, individually amplified by broadband amplifiers, and fed to the photonic processor. A signal generator was used to produce a tone with $f_{OLO} = 26$ GHz. Consequently, the processed signal is frequency down-converted to 2 GHz. The electrical output signal of the processor was sampled by a real-time sampling scope (RTSS), and processed offline. In all experiments performed in this work, offline processing involved only essential functions: frequency down-conversion to baseband, removal of the pilot tones through low-pass filtering, down-sampling to 1 sample per symbol, normalization and de-rotation of the constellation. Further details about offline signal processing can be found in the Supplementary Note 5. VOAs are responsible for power equalizing the four signals, and also for introducing a common extra attenuation.

The obtained results are displayed in Fig. 2c, d. As expected, the amplitude of the output signal increases proportionally to the number of enabled paths. However, the error vector magnitude (EVM) is more than halved when the number of enabled paths is doubled. Although the added signals are identical, these have different noise sources as these are amplified by different electrical and optical amplifiers. Therefore, coherently adding the signals increases the signal-to-noise ratio (SNR) of the output signal, thus explaining why its EVM is more than halved. These results validate the operation principle of the proposed photonic processor.

**Single-beam beamforming.** As depicted in Fig. 3a, demonstrating the proposed photonic processor as a photonic-aided payload receiver involves connecting the processor to a PAA receiver, and feeding multiple beams from different directions to the PAA. However, the processor requires adding pilot tones to the received signals. The simplest way of adding pilot tones to all signals is to

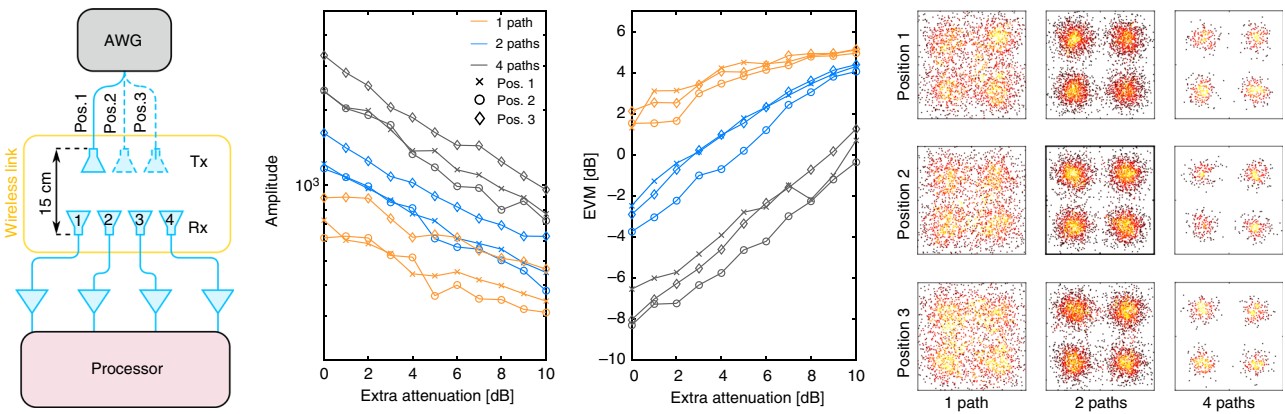

**Fig. 3** Demonstration of single-beam beamforming. **a** Set-up: the Tx antenna is placed at three different positions. **b** Amplitude and **c** error vector magnitude (EVM) of the output signal when adding one, two and all four signals, for the different positions of the Tx antenna. **d** Representative constellation diagrams of the output signal obtained without extra attenuation

use a dedicated antenna for transmitting the pilot tones to the entire PAA receiver. Such approach does not impact the low-noise amplification stage located right after the PAA receiver, as pilot tones are by definition much weaker than the input signals. The amplitude and phase of the pilot tones depend on the position of a given antenna element of the PAA receiver. Nonetheless, given that the position of the dedicated antenna is known beforehand, such a dependency can be calibrated and therefore compensated.

The simplified experimental set-up used for demonstrating single-beam beamforming is shown in Fig. 3a. For a matter of simplicity only a single channel of the AWG is used, meaning that beam and pilot tones are simultaneously generated and transmitted by a single Tx antenna. Due to power limitations, the Tx antenna is positioned 15 cm from the PAA receiver, which comprises four antenna elements uniformly spaced by 20.1 mm. The photonic processor was configured to coherently add the four input signals with equalized power and time delay. As a result, optimized beamforming is obtained without having to reconfigure the processor when changing the beam launch point. In order to validate such a statement, the performance of the system is assessed for three different position of the Tx antenna. As depicted in Fig. 3a, at position $k$ the Tx antenna is equidistant from the antenna elements $k$ and $k + 1$.

The results displayed in Fig. 3b–d show that beamforming was achieved for all positions of the Tx antenna. When taking as reference a given signal amplitude, the system provides a performance similar to the observed in Fig. 2c. Such allows to conclude that the introduction of a wireless link did not produce any impairment other than the unavoidable free-space path loss. Assuming that all four paths of the photonic processor provide identical performances, the output signal with lowest EVM should be obtained when the PAA receiver receives the maximum amount of power. Such case corresponds to placing the Tx antenna at position 2. A higher EVM should thus be obtained for positions 1 and 3. Both statements are confirmed by the experimental results. However, the output signal with the highest amplitude is not obtained when placing the Tx antenna at position 2, but at position 3. Such observation does not prove any inconsistency, as a signal with a higher amplitude does not necessarily have a higher SNR.

**Multi-beam beamforming**. The demonstration of a flexible photonic-aided payload receiver must assess its capability of separating multiple beams. In order to achieve so, the experimental set-up depicted in Fig. 4a is considered. An additional Tx antenna is connected to the second channel of the AWG for producing the

second beam. The AWG generates two identical RF signals with parameters as previously defined, but carrying distinct symbol patterns. Pilot tones are added only to one of the signals. Given that the photonic processor automatically points the receiver towards the direction from where the pilot tones originate, the processor automatically beamforms the beam to which pilot tones are added. Both Tx antennas should be as far as possible from the PAA receiver such that the angle of incidence of each beam is identical for all antenna elements of the PAA receiver. However, limited transmitting power forces placing the Tx antennas near the PAA receiver, at about 40 cm, even when using extra broadband amplifiers for boosting the transmitted power. Nonetheless, by setting the antenna $Tx_2$ 12 cm apart from the antenna $Tx_1$, the estimated power of the beamformed beam is 18.8 dB higher than the other (interfering) beam. The set-up thus poses no physical restriction to separating one beam from the other. Four cases are thus considered: beam 1 and pilot tones launched from $Tx_1$, $Tx_1(S)$, beam 1 and pilot tones launched from $Tx_1$ and beam 2 launched from $Tx_2$, $Tx_1(S) + Tx_2(I)$, beam 2 and pilot tones launched from $Tx_2$, $Tx_2(S)$, and beam 2 and pilot tones launched from $Tx_2$ and beam 1 launched from $Tx_1$, $Tx_2(S) + Tx_1(I)$.

The results displayed in Fig. 4b, c show once again that single-beam beamforming is achieved for both beams, regardless of the number of added signals. However, such is not the case when transmitting two beams. When a single path is enabled, the receiver is unable to distinguish one beam from the other. As both beams produce a similar RF power at any antenna element of the PAA, the photonic processor basically adds the two beams with identical weights, thus corrupting the beam intended to be beamformed. The activation of two paths enables six different combinations of pairs of antenna elements. Each combination results in a different nulling capability. Such explains why beam separation is achieved in some combinations (e.g. $1 + 2$ and $3 + 4$), and not in others (e.g. $1 + 3$ and $2 + 4$). The activation of all four paths provides clear beam separation as expected. The results also show that the performance of the various paths is not identical. Such a non-uniformity in performance in unavoidable, given that signals input to different paths are processed by different devices, such as electrical amplifiers, modulators and optical amplifiers. Nonetheless, such non-uniformity in performance is mitigated as more paths are enabled, and performance ends up being very similar when all four paths are enabled.

**Comparison with state-of-art OBFN**. Before discussing the obtained results, it is important to compare the proposed photonic-aided payload receiver with state-of-art OBFNs.

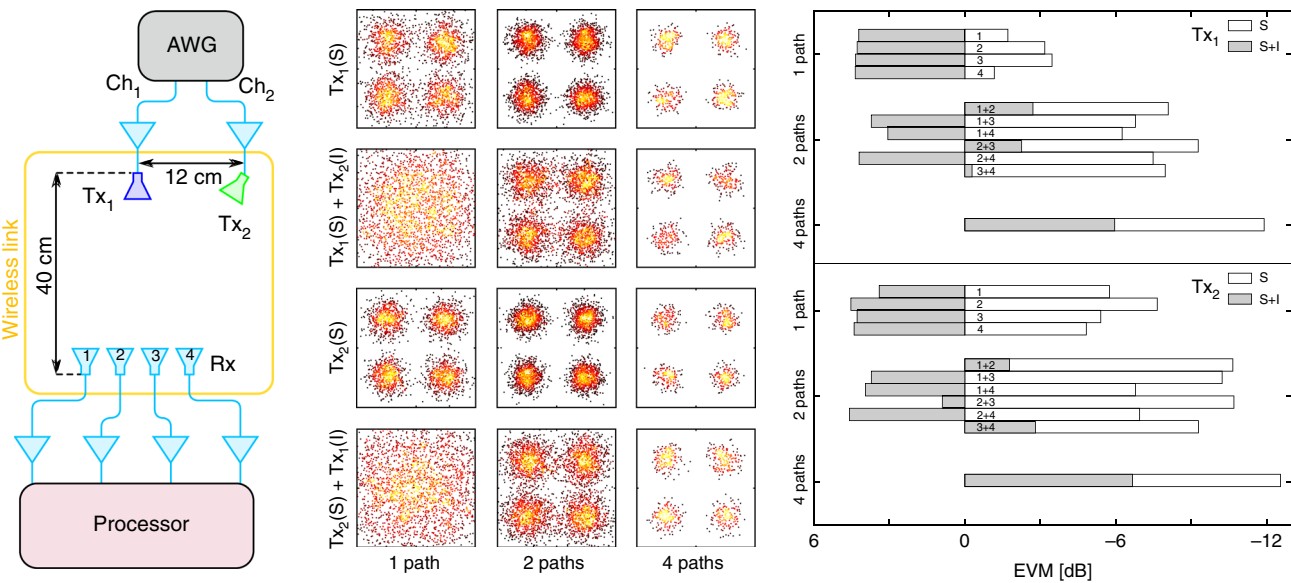

**Fig. 4** Demonstration of two-beam beamforming. **a** Set-up. **b** Representative constellation diagrams of the output signal obtained when one Tx antenna transmits the signal beam (*S*) and the other the interfering beam (*I*). **c** Error vector magnitude (EVM) of the output signal for different combinations of enabled paths and beams. All tests were made without introducing extra attenuation

The main motivation for photonic implementations of BFNs was, and still is, the fact that RF phase shifters are bulky, lossy and increasingly challenging to produce at high RF frequencies. Therefore, almost all of the research done in OBFNs has focused on developing photonic phase shifters, able to surpass such limitations. As explained in the following paragraph, the various photonic phase shifters can be divided in three generations.

The first-generation photonic phase shifters consist of mere optical fibres with a length tailored to provide the target RF phase shift[29]. Discretely adjustable phase shifting was demonstrated by switching among optical fibres with different lengths[30]. The second-generation photonic phase shifters exploit linear and nonlinear properties of photonic devices. Chromatic dispersion was the most exploited linear property[31–33]. Dispersive devices such as optical fibres and fibre Bragg gratings have a propagation delay that depends on the wavelength of the input signal. Such a property was exploited to produce continuously tunable photonic phase shifters in a single device. In terms of nonlinear properties, slow light was profoundly exploited as it enables modulating the refractive index of the medium with the input power[34–36]. Such an effect was used to induce a tunable phase shift between RF sidebands and optical carrier of an RF signal modulated onto an optical carrier. These phase shifters were the first not to resort to delay lines. The third-generation photonic phase shifters are based on adjustable optical filtering. TODLs based on all-pass filters implemented in resonant and non-resonant interferometers were, respectively, proposed in refs. [9,10,19,37,38]. Both filters have a periodic frequency response, which is tuned to provide the correct delay to at least one of the RF sidebands of the RF signal modulated onto an optical carrier. The TODLs used in the present work are based on non-resonant MZDIs. Its operation principle as well as its application to an OBFN are detailed in ref. [20]. Programmable filters based on liquid crystal on silicon (LCoS) matrices were also proposed both for providing a TODL and a phase shift between the optical carrier and a single RF sideband[39,40].

While a plethora of phase shifters and TODLs were proposed, few of these were experimentally demonstrated within an OBFN[10,21,30,32,41–43], especially within a receiving stage[10,41]. A receiving stage combines signals from the PAA to form one or multiple beams. Signal combination can be performed

coherently or incoherently. The latter option typically multiplexes signals into a wavelength-division multiplexing (WDM) signal, which is then directly detected[40,43,44]. Despite being a straightforward approach, it supports a limited number of channels, it does not resort to coherent detection, and consequently does not support heterodyne reception. Conversely, coherent signal combination does not have such drawbacks. In fact, coherent signal combination enables building an OBFN identical to an RF BFN, that is, signals are coherently combined without bandwidth limitations, RF frequency down-conversion is achieved by means of heterodyne reception, and optical phase shifting is equivalent to RF phase shifting[25,28]. The latter advantage enables the use of the simplest and smallest photonic phase shifter—the optical phase shifter. However, to the best of the authors' knowledge, an OBFN resorting to coherent signal combination was demonstrated only once[10], without any active stabilization loop.

A comparison between the proposed system and state-of-art OBFNs would be desirable concerning key metrics such as size, weight, power consumption and performance degradation. However, such a comparison cannot be made as other proposed systems either lack dimensioning or were not dimensioned for communications satellites. Nonetheless, the proposed system is evaluated in detail according to such key metrics in ref. [5].

## Discussion

Future HTCSs will have to do better than just increasing capacity: they must become capable of providing flexible coverage for best serving fast-evolving user needs. Both requirements can only be achieved by adopting a reconfigurable antenna—a PAA—which in turn requires a massive signal processor, the BFN. While different implementations of the BFN have been discussed—analogue, digital or photonic-aided—the unique miniaturization and low-loss capabilities enabled by the latter puts photonics in a leading position. Nonetheless, a viable photonic-aided payload must comply with key features such as being scalable, modular, miniaturizable and having an architecture resembling an RF payload. Devising a simple, scalable and low-speed stabilization loop has proved to be the last challenge to overcome before such a photonic-aided payload could be demonstrated. The work

presented in this paper shows that such a challenge was indeed overcome, paving the way for a photonic revolution in HTCS payloads.

Scaling the proposed system to more antenna elements and more beams depends on the scalability of individual modules. The photonic BFN is both the most important and most challenging module to scale, as its complexity is proportional to both the number of antenna elements and number of beams. On the one hand, self-heterodyne detection makes its basic elements—phase shifters and couplers—equivalent to their RF counterparts, thus enabling the miniaturization of the BFN by a factor of $\lambda_{\mathrm{opt}}/\lambda_{\mathrm{RF}} \approx 5000$[5]. On the other hand, as the number of required phase shifters is proportional to the number of antenna elements of the PAA and to the number of beams, about 25,000 phase shifters are required[5]. Therefore, the development roadmap of a high-capacity and flexible photonic-aided payload should focus on developing dense arrays of low-speed phase shifters with low power consumption and low insertion losses.

Even though the focus of the present work is the real-time demonstration of a modular and coherent photonic-aided payload receiver, such a demonstration included several novelties worth highlighting: first-ever use of a EDMCFA in an OBFN, first-ever integrated OBFN including a photodiode, first-ever demonstration of an OBFN performing RF frequency conversion by means of heterodyne reception and first-ever separation of two beams by an OBFN.

A discussion of future directions should be split in two parts: system and devices. From a system perspective, although challenging, scaling the proposed receiver to more antenna elements and more beams does not require deep modifications to the basic architecture and stages presented in Fig. 2a. Nonetheless, although the proposed system is based on a single laser source, multiple laser sources with different wavelengths can be used where suitable, as the proposed system is compatible with WDM. Adapting the proposed architecture to the transmitting stage would be straightforward, also requiring a simpler MCL, as $N$-to-1 signal combination becomes signal splitting.

From a device perspective, we believe that a significant progress in PICs is mandatory such that beamforming one beam in a single PIC can be envisaged. Focus should be given to developing low-loss interfacing as well as low-loss, compact, low-power phase shifters.

The method used for interfacing depends on whether WDM is used. If not, as considered in the present work, it requires $N + 1$ input fibres, provided that polarization multiplexing is also not used. Such a large number of fibres should be bundled into a two-dimensional fibre array, which should be precisely aligned with a set of grating couplers. While such interfaces with these many inputs/outputs have already been investigated[45], developing a space-qualified package requires a significant effort. Using WDM means that only a few interfacing fibres are required, which does not pose a problem in interfacing. However, an OLO must be allocated for each channel, and all resulting signals have to be demultiplexed on chip, which requires large and lossy demultiplexers[43]. It is therefore likely that a trade-off between both solutions should prove to be the best solution.

As discussed in ref. [28], silicon PICs offer three kinds of phase shifters based on different effects: thermo-optic, carrier injection and carrier depletion. These phase shifters have different trade-offs among insertion loss, voltage-dependent loss, footprint and power consumption, all being unacceptably underperforming in at least one of such parameters. For instance, thermo-optic and carrier-injection phase shifters consume more than 10 mW[28], which according to the model presented in[5] puts the power consumption of phase shifters at the same level of the low-noise amplifiers. Conversely, carrier-depletion phase shifters have

negligible power consumption, but at the cost of being almost 1 cm long and lossy, with over 5 dB of insertion loss. Consequently, a new kind of phase shifter should be developed. Liquid crystals are known to have a very strong electro-optic effect, orders of magnitude higher than materials widely used in photonics such as lithium niobate, enabling very small phase shifters[46,47]. They are also very transparent to light, and thus low-loss, and are routinely integrated in LCoS matrices nowadays packing over $10^7$ phase shifters in <1 cm$^2$ [48]. Co-integrating a reflective LCoS matrix on top of silicon PIC using grating couplers to provide vertical interfacing appears to be a promising approach, allowing to envisage a phase shifter as large as a grating coupler, typically with $10 \times 10$ μm$^2$ [46].

Liquid crystals have a slow response on the order of 1 ms. Although such response is fast enough for compensating phase wandering, it might be insufficient for the modulation of dithering tones at frequencies higher than 1 kHz. Nonetheless, such is not a problem. Even though the number of required phase shifters $\beta$ is of $N \times N_{\mathrm{B}}$, the number of required dithering tones is of only $N$. Consequently, only $N$ modulators are required to produce the dithering tones. In order to avoid parasitic phase dithering of the RF signal associated with the modulation of the dithering tones, a suitable modulation scheme would be to resort to a ring modulator only modulating one of the pilot tones[49]. While such a modulation scheme provides intensity modulation instead of phase modulation, the operation principle behind the MCL would be the same.

The proposed coherent photonic processor can be generalized from an $N$-to-1 weighted adder to a $M \times N$ matrix multiplier without changing the MCL. Optical matrix multipliers resorting to coherent optical signal processing have been proposed as a promising alternative approach to microelectronic and optoelectronic artificial neural networks, as once the matrix coefficients are set the multiplication takes place at the speed of light[50]. As a single PIC can pack a very limited number of neurons, implementing larger and more powerful networks requires multiple PICs, and therefore modularity. Similarly to the proposed processor, scaling up to a modular implementation must conserve coherency between all signals in all modules, also requiring an MCL. Therefore, the presented work may serve as a starting point towards a modular coherent photonic neural network.

## Methods

**Monitoring and control loop.** As explained in the main text, the MCL sets and stabilizes the amplitude, phase and delay of each input signal. Here, we detail the operation of the MCL.

The MCL first sets the TODLs as described in ref. [28] to minimize the relative delay among signals. Given that the configured delays remain stable over time, for a matter of simplicity the MCL configures one TODL at a time, with dithering tones deactivated. In order to avoid interference from other paths, all paths except for the one including the TODL being configured are fully attenuated.

Once the TODLs are configured, the power of all signals is equalized as follows. The power of each signal is measured one at a time, with all other signals fully attenuated to avoid any measurement error. The power of the weakest signal is then subtracted a margin of 1 dB, resulting in a reference power $P_{\mathrm{ref}}$. Each signal is attenuated until its power falls within $P_{\mathrm{ref}} \pm P_{\mathrm{thres}}$, where $P_{\mathrm{thres}} = 0.1$ dB. At this point the extra attenuation mentioned in the figures is of 0 dB.

After all signals are again attenuated until the target extra attenuation is reached, the system is ready to coherently combine all signals. The MCL is then configured to proceed similarly to a phase-locked loop, with each iteration described as follows. Dithering tones are simultaneously generated with a default amplitude of 50 mV, and with a frequency of $f_{\mathrm{d},k} = 500(k + 1)$. The DSC then configures the ADC to capture 128 samples at 64 ksa s$^{-1}$. The dithering tones are deactivated once all samples have been captured. The delay between the generation of the dithering tones and the sampling by the ADC was measured to be fairly constant, with a jitter of only a few. The samples produced by the ADC are processed by the DSC to extract the dithering tones with the highest frequency, that is, $f_2 + f_{\mathrm{d},k}$, from which the phase of each optical signal is estimated. Even though the estimated phases have an undetermined phase reference, such a phase reference is the same for all estimated phases. As a result, all paths have the same phase if all estimated phases are identical, regardless of the common value. The objective of the

MCL thus is how to achieve so by producing the smallest possible ajustment to the phase shifters $\beta$. Given that voltage adjustments are proportional to estimated phases, the solution is to minimize the variance and average value of the estimated phases. Such is performed as follows. A vector containing the estimated phases is first sent to the computer. Variance is minimized by adding or subtracting $2\pi$ to each estimated phase; if the resulting variance is reduced, the estimated phase is updated. Once the variance has been minimized, the average of the vector containing the updated phases is set to zero. The voltage offset to be applied to a given phase shifter is varied by the estimated phase multiplied by a gain factor, which by default is $-0.2\text{V}/\pi$. If the new voltage offset is higher than $V_{\max} = -0.7$ V or lower than $V_{\min} = -1.5$ V, it is reset to $(V_{\max} + V_{\min})/2 = -1.1$ V. This is necessary to guarantee that the phase shifters $\beta$ operate within voltage limits. The new voltage offsets are transmitted to the DSC, which updates the DACs associated with the corresponding phase shifters $\beta$. The measured duration of each iteration and thus the period of the MCL was of 100 ms, dominated by the communication time between the DSC and the computer, performed via USB.

## Data availability

The data that support the findings of this study are available from the corresponding author upon reasonable request.

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

## Acknowledgements

This work was supported in part by the Fundação para a Ciência e Tecnologia/Ministério da Educação e Ciência under the Ph.D. Grant SFRH/BD/117444/2016, by Fundo Europeu de Desenvolvimento Regional–Portugal 2020 partnership agreement under the project UID/EEA/50008/2013 and by the European Commission through the project BEACON (FP7-SPACE-2013-1-607401).

## Author contributions

M.V.D., R.N.N. and V.C.D. jointly developed the concept. V.C.D., G.W. and L.Z. designed and fabricated the chip. J.G.P. and C.F.R. implemented the MCL and designed the layout of the PCB. R.W. and S.C. conceived the array of MZMs. M.F., M.N. and T.N. conceived the radiation-hard EDMCFA fibre and fan-in/fan-out. J.C., L.S. and M.K. assembled the EDMCFA. J.A. coordinated and gave the inputs for space applications. V.C.D. and M.V.D. conceived the experiments, performed the measurements and analysed the data. V.C.D, M.V.D. and R.N.N. wrote the paper. M.V.D., S.C. and R.N.N. managed and coordinated the project.

## Additional information

**Competing interests:** The authors filed a patent application on a photonic system to perform beamforming of a radio signal: Miguel V. Drummond, Rogério N. Nogueira and Vanessa C. Duarte, Photonic beamforming system for a phased array antenna receiver, PCT/IB2016/052206, led on 18 April 2016. The remaining authors declare no competing interests.

