## [Peer Review File · Nature Communications]

This manuscript has been previously reviewed at another journal that is not operating a transparent peer review scheme. This document only contains reviewer comments and rebuttal letters for versions considered at Nature Communications. Mentions of the other journal have been redacted.

Reviewers' Comments:

Reviewer #1:

Remarks to the Author:

I have checked with interest the revisions and replies made by the authors and agree with most of them. However I still need to be convinced that the solution they propose for handling the very high number of phase shifters is compatible with space applications. What I mean with this is the impact that harsh environmental and mechanical conditions would have over the liquid crystal matrices on silicon and the in-going and out-going fiber array. I doubt this approach would stand space qualification tests.

Reviewer #2:

Remarks to the Author:

The manuscript presents the architecture and the proof-of-concept implementation of a photonic beamforming system for HTCS. Several novel achievements are claimed, the most important of which being the first demonstration of a receiving BFN capable of focusing towards different input beams.

The results described in the paper demonstrate the feasibility of the PBS, and discusses its suitability to realistic antenna arrays.

The work is convincing and novel, and has a moderate potential to influence its field.

Reviewer #3:

Remarks to the Author:

The authors have well answered the questions raised by myself and the other reviewers. However, I think the manuscript still needs some additional revision before it can be considered for publication in Nature Communications.

1) The authors also agree that the entire system is a photonic-aided payload "receiver" rather than a satellite payload. I think the title of the manuscript should be modified accordingly. "Modular coherent photonic-aided payload receiver for communications satellites" might be better.

2) I strongly suggest that "PBS" not be used as the abbreviation of "photonic beamforming system" because PBS is widely known as "polarization beam splitter" for people working in photonics.

3) How many symbols did the authors use for the demodulation of 4QAM data in the experiment? As can be seen from the Fig. 2 (d), Fig. 3(d) and Fig. 4(b), the symbols used for the "4 paths" results seems less than that of the "1 path" and "2 path" results.

4) The authors may explain why phase shifters rather than true time delay elements are used for photonic beamforming? True time delay is a better solution for beamforming as it can avoid the problem of beam squint.

Reviewer 1

I have checked with interest the revisions and replies made by the authors and agree with most of them. However, I still need to be convinced that the solution they propose for handling the very high number of phase shifters is compatible with space applications. What I mean with this is the impact that harsh environmental and mechanical conditions would have over the liquid crystal matrices on silicon and the in-going and out-going fibre array. I doubt this approach would stand space qualification tests.

Reply to reviewer 1

The question of whether liquid crystal based technologies (such as LCoS) are able to endure the harsh environmental and mechanical conditions related with satellite launching and orbiting is extremely pertinent.

We did a thorough search for Space qualification test results involving liquid crystal technologies. Even though we could not find a complete Space qualification of any device, we did find a very recent work by Manuel Silva-López, Antonio Campos-Jara, Alberto Álvarez Herrero from <http://www.inta.es>, “Validation of a spatial light modulator for space applications”, paper P54, presented at ICSO 2018 (www.icso2018.org). This is probably the state of art of space qualification of LCoS. In a nutshell, even though tests are preliminary, the tested LCoS operated well and without degradation in vacuum and thermal tests, just needing a better calibration for such conditions. The authors conclude that “These preliminary results show that this particular LCoS model may be suitable for space applications.”, but, indeed, much more testing is needed before strong conclusions can be drawn.

As for the fiber array, we could not find any recent work on space qualification. However, we are aware that many photonic devices such as lasers, modulators and photodiodes that naturally resort fiber coupling have been space qualified or are close of being so. This makes us optimistic that qualifying fiber arrays should not be barred by unsurmountable problems.

All in all, even though space qualification is a long road, the most recent test results are encouraging.

Reviewer 2

The manuscript presents the architecture and the proof-of-concept implementation of a photonic beamforming system for HTCS. Several novel achievements are claimed, the most important of which being the first demonstration of a receiving BFN capable of focusing towards different input beams. The results described in the paper demonstrate the feasibility of the PBS, and discusses its suitability to realistic antenna arrays.

The work is convincing and novel, and has a moderate potential to influence its field.

Reply to reviewer 2

Thank you for your kinds words.

Reviewer 3

The authors have well answered the questions raised by myself and the other reviewers. However, I think the manuscript still needs some additional revision before it can be considered for publication in Nature Communications.

1. The authors also agree that the entire system is a photonic-aided payload “receiver” rather than a satellite payload. I think the title of the manuscript should be modified accordingly. “Modular coherent photonic-aided payload receiver for communications satellites” might be better.
2. I strongly suggest that “PBS” not be used as the abbreviation of “photonic beamforming system” because PBS is widely known as “polarization beam splitter” for people working in photonics.
3. How many symbols did the authors use for the demodulation of 4QAM data in the experiment? As can be seen from the Fig. 2 (d), Fig. 3(d) and Fig. 4(b), the symbols used for the “4 paths” results seem less than that of the “1 path” and “2 path” results.
4. The authors may explain why phase shifters rather than true time delay elements are used for photonic beamforming? True time delay is a better solution for beamforming as it can avoid the problem of beam squint.

Reply to reviewer 3

Comment 1

The authors also agree that the entire system is a photonic-aided payload “receiver” rather than a satellite payload. I think the title of the manuscript should be modified accordingly. “Modular coherent photonic-aided payload receiver for communications satellites” might be better.

Reply

We agree, and changed the manuscript accordingly.

Comment 2

I strongly suggest that “PBS” not be used as the abbreviation of “photonic beamforming system” because PBS is widely known as “polarization beam splitter” for people working in photonics.

Reply

We agree, and replaced PBS by OBFN (optical beamforming network).

Comment 3

How many symbols did the authors use for the demodulation of 4QAM data in the experiment? As can be seen from the Fig. 2 (d), Fig. 3(d) and Fig. 4(b), the symbols used for the “4 paths” results seem less than that of the “1 path” and “2 path” results.

Reply

Such information is detailed in the Supplementary Information, section “Offline digital signal processing”. We point out to the following sentences:

- “the RTSS outputs a frame that is 640 symbols long, i.e., as long as the frame produced by the AWG.” We therefore demodulate an entire frame, 640 symbols long.
- “The presented values of amplitude, EVM and SER are the average of 100 frames.”

- *“The representative constellations are the overlap of the constellations of the best frames for every possible combination, i.e., paths 1, 2, 3 and 4 for one enabled path; paths 1 + 2, 2 + 3, 3 + 4, 1 + 3, 2 + 4 and 1 + 4 for two enabled paths; and all four enabled paths.”*

As a result of the last pointed out sentence, your observation is correct. For 4 paths, there is only one constellation, for 2 paths there are 6 overlapped constellations, and for 1 path there are 4 overlapped constellations.

Given that the requested information is already present in the manuscript, no changes were made.

Comment 4

The authors may explain why phase shifters rather than true time delay elements are used for photonic beamforming? True time delay is a better solution for beamforming as it can avoid the problem of beam squint.

Reply

Excellent point. Even though we deliberately wrote few words about time delay, and never mention “true-time delay”, the demonstrated beamforming was indeed true-time delay. This is clearly stated in section “Single-beam beamforming”, 2nd paragraph: *“The photonic processor was configured to coherently add the four input signals with equalized power and time delay.”*

The reasons why we wrote few words about true time delay are the following:

1. In a coherent OBFN, using phase shifters is mandatory to enable coherent combination of signals. Conversely, using true-time delay is optional: it should only be used if beam squinting is severe, as otherwise it'll bring nothing but more complexity and especially size;
2. We already provided an extensive analysis on true time delay beamforming in reference [20];
3. We already demonstrated true time delay elements (and corresponding control loop) in reference [28];
4. We observed in reference [5] that most beams, if not all, are squint-free, thus dispensing true-time delay beamforming;
5. Finally, the demonstrated true time delay elements have a very small tuning range of 50 ps, which was chosen very early in the project for purely demonstration purposes. A beamformer should be named as true-time delay only if the delays are higher than a symbol period (in the demonstrated case, of 1 ns).

These points are summarized in the Supplementary Information, last paragraph of page 4. As a result, no changes were made to the manuscript.